# Assessing the extinction risk of the spontaneous flora in urban tree bases

**Apolline Louvet**[1]�*, **Clément Mantoux**[2]�, **Nathalie Machon**[3]

**1** Department of Mathematical Sciences, University of Bath, Bath, United Kingdom, **2** INRIA Paris, Paris, France, **3** Centre d'Ecologie et des Sciences de la Conservation, Muséum National d'Histoire Naturelle, Sorbonne Université and Centre National de la Recherche Scientifique, Paris, France

055 These authors contributed equally to this work.
\* apolline.louvet@polytechnique.edu

**Data Availability Statement:** The data supporting the findings of this study are openly available on Zenodo at https://doi.org/10.5281/zenodo.3770339. The codes used as part of this study are

## Abstract

As the spatial arrangement of trees planted along streets in cities makes their bases potential ecological corridors for the flora, urban tree bases may be a key contributor to the overall connectivity of the urban ecosystem. However, these tree bases are also a highly fragmented environment in which extinctions are frequent. The goal of this study was to assess the plant species' ability to survive and spread through urban tree bases. To do so, we developed a Bayesian framework to assess the extinction risk of a plant metapopulation using presence/absence data, assuming that the occupancy dynamics was described by a Hidden Markov Model. The novelty of our approach is to take into account the combined effect of low-distance dispersal and the potential presence of a seed bank on the extinction risk. We introduced a metric of the extinction risk and examined its performance over a wide range of metapopulation parameters. We applied our framework to yearly floristic inventories carried out in 1324 tree bases in Paris, France. While local extinction risks were generally high, extinction risks at the street scale varied greatly from one species to another. We identified 10 plant species that could survive and spread through urban tree bases, and three plant traits correlated with the extinction risk at the metapopulation scale: the maximal height, and the beginning and end of the flowering period. Our results suggest that some plant species can use urban tree bases as ecological corridors despite high local extinction risks by forming a seed bank. We also identified other plant traits correlated with the ability to survive in tree bases, related to the action of gardeners. Moreover, our findings demonstrate that our Bayesian estimation framework based on percolation theory has the potential to be extended to more general metapopulations.

## Author summary

Understanding how biodiversity is maintained in a urban environment is an important question in ecology. In this article, we investigated to what extent tree bases along streets contribute to maintaining biodiversity in cities by acting as ecological corridors between larger urban green spaces. To do so, we introduced a new estimation framework to assess whether a plant species can survive and spread through urban tree bases, given presence/

openly available on GitHub at https://github.com/cmantoux/boa-process.

**Funding:** This work was supported by ANR (Agence Nationale de la Recherche) Écoville [ANR 14 CE22-0021]. CM acknowledges support from the "Investissements d'avenir" program (PRAIRIE 3IA Institute, [ANR-19-P3IA-0001]) and the European Research Council [SEQUOIA-724063]). AL acknowledges support from the Royal Society, and partial support from the chair program "Mathematical Modelling and Biodiversity" of Veolia Environment-Ecole Polytechnique-National Museum of Natural History-Foundation X. The funders did not play any role in the study design, data collection and analysis, decision to publish, or preparation of the manuscript.

absence data (such as floristic inventories). The novelty of our approach is to combine two factors previously identified as particularly relevant in a urban environment: selection against dispersion, and the potential presence of a seed bank of dormant seeds in the soil. We applied our estimation framework to a dataset of yearly floristic inventories carried out in 1324 tree bases in Paris, France. We identified several plant traits making it possible to use urban tree bases as ecological corridors. Our study sheds light on how urban tree bases are integrated into the urban ecosystem. Our findings can be used to design improved management strategies for urban tree bases, that will contribute to biodiversity preservation in urban environments.

## Introduction

Understanding how species survive in fragmented landscapes is an important issue in ecology [1]. Of particular interest is understanding how plant species persist in urban environments. Indeed, while highly fragmented and subject to frequent external perturbations [2], urban environments host a wide variety of vascular as well as exotic plant species [3, 4], and are richer than rural areas of the same size from a biodiversity viewpoint.

Urban environments are often composed of patches with very different features, which can broadly be divided into two categories: large urban green spaces (such as parks or gardens), in which perturbations are less frequent and populations can grow large enough to avoid extinction due to stochastic fluctuations, and small patches (such as tree bases found along streets, small lawns, or fragmented railway environments), in which survival is the result of a balance between local extinctions and colonisation [5, 6]. The deliberate spatial arrangement of these smaller patches makes them potential ecological corridors between larger urban green spaces (as already evidenced for some species, see [5, 7, 8]), increasing landscape connectivity and gene flow between larger urban green spaces. Therefore, they may contribute to the overall viability of plant populations in urban environments. As previous studies showed that low-range dispersal could be favoured against long-range dispersal in this highly fragmented environment [9], the mechanisms that ensure survival in such an environment are expected to be different from the ones generally at play in metapopulations. In this study, we focus on a specific type of urban environment: the base of the trees planted along streets in cities (see Fig 1 for an illustration), and aim at assessing to what extent urban tree bases contribute to the overall connectivity of the urban ecosystem and to the viability of plant metapopulations in cities. Indeed, tree bases are occupied by a spontaneous flora (that is, plants that were not intentionally introduced and cultivated by gardeners, also referred to as "weeds"), whose dynamics has been the subject of growing interest over the past decade [5, 8, 10, 11]. Recent work identified a first plant trait that improves survival in urban tree bases: the ability of seeds to stay dormant within the soil during several generations without losing viability, or in other words, the ability to disperse in *time* rather than in space by forming a seed bank [8, 11]. The main goals of this article are 1) to assess which species are able to survive and spread through urban tree bases given the potential presence of a seed bank in the soil, 2) to study whether the formation of a seed bank is necessary for survival in urban tree bases, and 3) to uncover other biological traits improving survival in such an environment. To do so, we used a dataset of annual floristic inventories of the spontaneous flora found in urban tree bases, carried out on 1324 tree bases located in Paris, France from 2009 to 2018 (the *Paris 12* dataset, [12]).

In order to study colonization and extinction dynamics given presence/absence data, a classical modelling approach is to use a family of models called *Stochastic Patch Occupancy Models*,

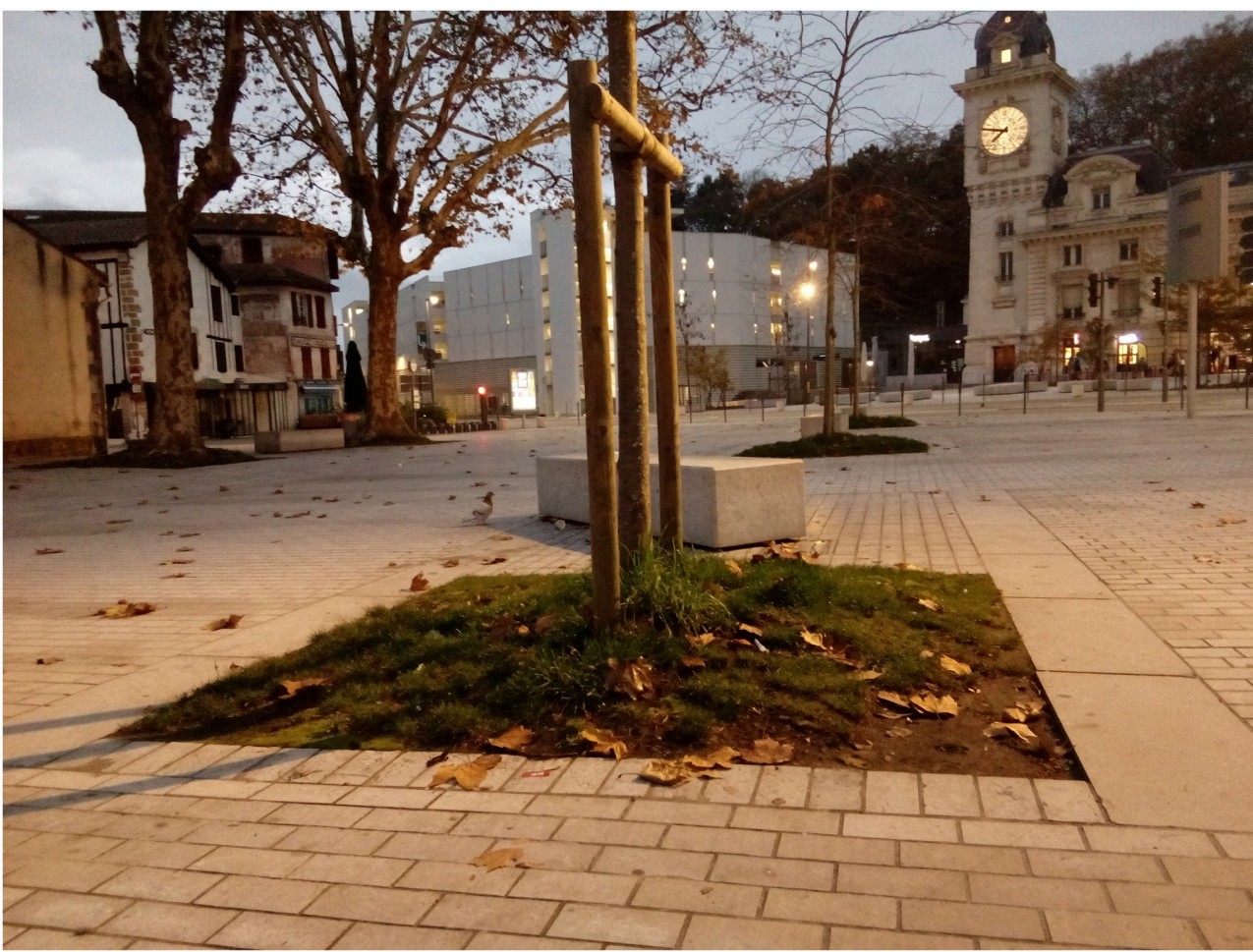

**Fig 1. Example of urban tree bases with spontaneous flora.**

or SPOMs (see e.g. [13] for an overview). As these models only require one or several snapshots of patch occupancies, and do not require demographic data, they have been used extensively to study metapopulations in fragmented landscapes and answer questions related to species survival arising in conservation biology (see [14] for a recent review). While a wide variety of SPOMs exist [13, 14], they generally do not take into account the potential presence of a soil seed bank, and are hence ill-suited to plant metapopulations [15, 16]. Therefore, extending this modelling framework to include dormancy is an active area of research in metapopulation theory, and new SPOMs with a seed bank component have been recently introduced [17, 18]. In the case of [17], associated statistical tools have been developed, and allow for statistical inference using presence/absence data [11, 17, 19]. However, this model assumes the presence of a constant influx of seeds from an external source, which does not make it possible to observe metapopulation extinctions, and is hence not suited to the study of extinction risks. Conversely, in the model introduced in [18], called the *Best Occupancy Achievable* process (or BOA process), colonization only occurs between direct neighbours, and the conditions under which the metapopulation has a high extinction risk have been identified [18]. In this article, we develop inference tools for the BOA process, and apply them to urban tree bases to assess the extinction risk of the spontaneous flora. To do so, we 1) develop a Bayesian

framework for parameter inference under a BOA process, and 2) introduce a new metric measuring the extinction risk of plant metapopulations in the absence of external colonisation, given the results of the estimation procedure. We then 3) apply the estimation framework to the *Paris 12* dataset, to assess which species can survive and spread through urban tree bases without external colonisation (low value of the extinction risk metric) and which species cannot survive without a constant supply of propagules from outside the urban tree bases (high value of the extinction risk metric). We conclude by 4) interpreting these results in light of the traits of the plant species, and by 5) discussing how our estimation framework of the extinction risk can be extended to more general metapopulation models.

## Materials and methods

### Study area

In this study, we used a dataset of floristic inventories of the spontaneous flora in 1324 tree bases located in Paris 12th administrative district [12]. An exhaustive inventory was carried out annually from 2009 to 2018, except in 2013 where a limited number of species were tracked (see Table B in S1 Text for details on the nomenclature used and the list of species tracked in 2013). The taxonomic reference used is the French Flora Reference TAXREF v8.0 [20].

Throughout this study, we assumed that colonisation from an external source was negligible compared to colonisation between neighbouring tree bases. Moreover, following evidence of selection for low-distance dispersion in an urban environment [9], we assumed that colonisation was only possible between direct close neighbours. Therefore, we divided each street into portions of contiguous and approximately equidistant tree bases.

For each portion of street listed in Table A in S1 Text, we focused on the species observed in an average of at least 10% of its tree bases over the time period 2009–2018 for species tracked in 2013, and over the time period 2014–2018 for the others (see Table B in S1 Text). We assumed that each resulting pair of species and portion of street corresponded to an independent metapopulation. In total, we kept for analysis a total of 181 pairs of plant species and portion of street: 133 for the time period 2009–2018, and 48 for the time period 2014–2018. The complete list of pairs can be found in Table B in S1 Text. For each pair of plant species and portion of street, our goal was to assess whether the species was at risk of going extinct in this portion of street in the absence of external colonisation. This required determining whether a seed bank was present in the soil given the observations of standing vegetation and, if so, estimating how long the seeds could stay dormant.

### Model used

The model we used in this study is a variant with a seed bank component of the Levins model [6]. This model, called the *Best Occupancy Achievable* process (or BOA process), was introduced in [18] as the large seed production limit of an individual-based population genetics process describing plant dynamics in a metapopulation. The BOA process belongs to the family of Stochastic Patch Occupancy Models (or SPOMs, see e.g. [13] for a general introduction), and encodes whether plants and seeds are present or absent in each patch of a metapopulation whose patches are equidistant and distributed along a line. More precisely, the model is characterised by the following two parameters:

- The *patch extinction probability* $p_{ext}$, which is the probability that all the plants in a patch go extinct before reaching the seed production stage.

- The *maximal dormancy duration H*, which is the maximum number of complete generations that seeds can remain dormant without losing viability.

The initial proportion $s$ of patches containing viable seeds can be considered as an extra parameter of the model.

The model evolves as follows: 1) At the beginning of each generation, we know whether each patch contains viable seeds, and the age of these seeds. For each patch, if it contains seeds, then some of them germinate and grow into plants. The remaining ones age and lose viability if they were produced more than $H + 1$ generations ago. 2) Each patch is affected by an extinction event with probability $p_{ext}$, independently of the other patches. If such an event occurs, all the plants contained in the patch die, while the seed bank is unaffected. We then record which patches still contain plants and which patches are empty. 3) At the end of the generation, before dying, the remaining plants produce new seeds which enter the seed bank of the patch and of the two neighbouring patches. See Fig 2 for an illustration of the model dynamics and see S2 Text (Section 1) for a formal definition of the BOA process as a Markov process.

As shown in [18], the main feature of the BOA process is the existence of a *critical patch extinction probability* $p_c(H)$ depending only on the maximal dormancy duration such that:

- If $p_{ext} > p_c(H)$, the metapopulation goes extinct in finite time (almost surely).

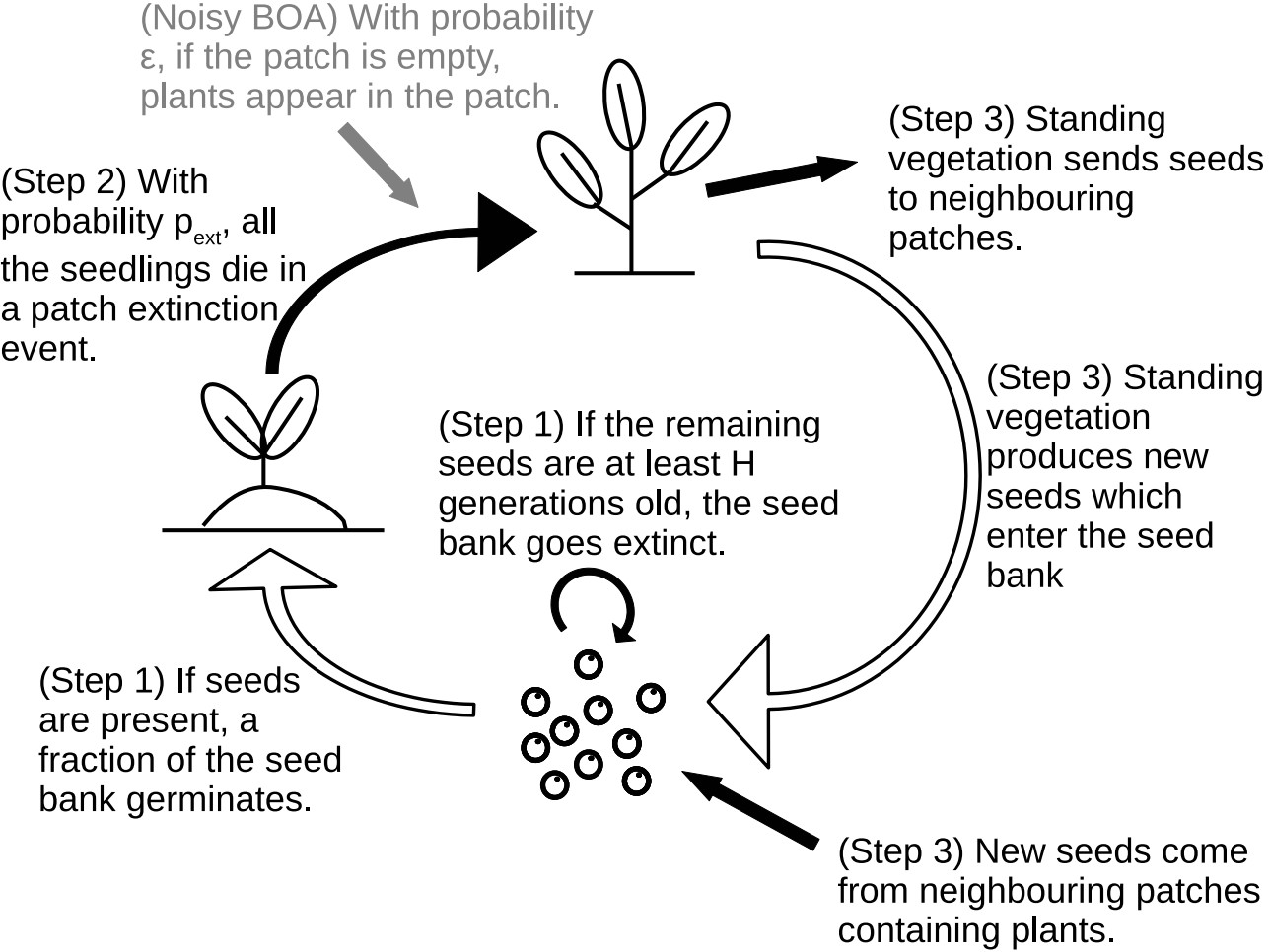

**Fig 2. Illustration of the dynamics of the BOA process with *patch extinction probability* $p_{ext} \in [0, 1]$ and *maximal dormancy duration* $H \in \mathbb{N}$, and its variant with a *noise parameter* $\epsilon \in [0, 1]$.** The parts in grey are those specific to the noisy BOA process variant.

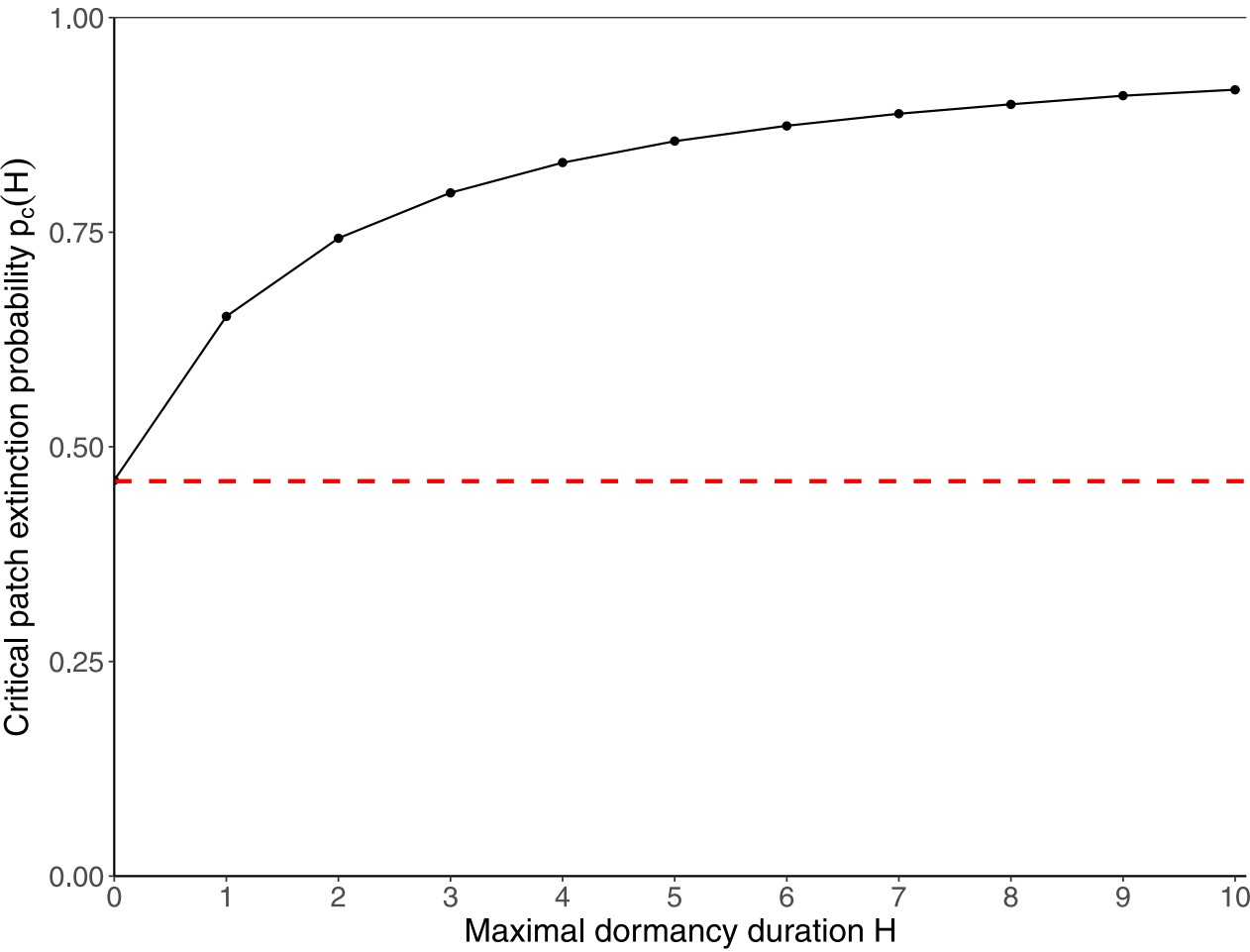

**Fig 3. Evolution of the *critical patch extinction probability* $p_c(H)$ for the BOA process, as a function of the *maximal dormancy duration H*.** The red dotted line indicates the critical value $p_c(0)$ above which survival at the metapopulation scale in the BOA process is not possible in the absence of a seed bank. The thin black line indicates the limiting value of $p_c(H)$ when $H \to +\infty$. Figure adapted from [18], with improved estimates for $p_c(H)$.

- If $p_{ext} < p_c(H)$, the metapopulation survives and spreads with nonzero probability, and the survival probability increases with $|p_{ext} - p_c(H)|$ (though the exact survival probability is unknown and is likely to strongly depend on the initial condition). Moreover, survival is made possible by the metapopulation spreading through previously unoccupied patches.

See Fig 3 for an illustration of the dependence of the critical patch extinction probability $p_c(H)$ on $H$. Our goal was to make use of this property of the BOA process to measure the extinction risk, by estimating $p_{ext}$ and $H$, and comparing the estimation with $p_c(H)$. As survival is equivalent to spreading through the set of patches, we can then interpret a low extinction risk as indicative of the ability of the metapopulation to use these patches as ecological corridors.

## Estimation method

The estimation method described in this section was implemented as part of a package available online at https://github.com/cmantoux/boa-process.

The BOA process can be considered as a Hidden Markov Model [21, 22]: we only observe the presence/absence of plants in each patch (after the patch extinction step but before the seed production step), while the state of the seed bank is a hidden state which influences the observations. In this study, we propose to use a Bayesian framework to estimate the model parameters. Indeed, given the complex hierarchical structure of the model, classical Maximum Likelihood estimation methods can only give the most likely value for $s$, $p_{ext}$ and $H$. In particular, they cannot be used to obtain confidence intervals and compare $p_{ext}$ to $p_c(H)$. Therefore, we propose to determine the posterior distribution $\mathbb{P}(s, p_{ext}, H|\text{Obs})$ of the parameters $s$, $p_{ext}$ and $H$ given the yearly observations of the standing vegetation. Since we only have 10 years of observations, we introduced an upper bound $H_{max} = 10$ on the value of $H$ to avoid identifiability issues. See S2 Text (Section 2) for more details on the Bayesian methodology used.

Intuitively, we can expect the estimator to identify easily when $H > 0$. Indeed, the observation of plants in a patch that was empty along with its two neighbours in the previous generation is characteristic of the presence of a seed bank, since such an event has probability zero if $H = 0$. As the probability that the patch and both of its neighbours were all affected by an extinction event during the previous generation is equal to $p_{ext}^3$, we expect to observe at least one such characteristic event with high probability in datasets of $\sim 50$ patches monitored during 5 or 10 years. We can also define similar characteristic events for $H \geq H_0$, $H_0 \in \mathbb{N} \setminus \{0, 1\}$, but they occur in each patch with probability $p_{ext}^{3H_0} << p_{ext}^3$, so we generally do not observe them in datasets of the size we consider in this study. Therefore, unless the extinction probabilities are very large, it is likely that the estimator cannot accurately differentiate medium dormancy durations from long maximal dormancy durations (however, note that the distribution of the age of seeds in the initial condition was chosen in such a way that there are no identifiability issues, see S2 Text). In particular, this implies that the upper bound introduced earlier on the value of $H$ does not affect the quality of the estimation, since dormancy durations larger than $H_{max}$ generally cannot be differentiated from medium dormancy durations.

The possibility of events characterising the presence of a seed bank implies that the estimation method is expected to be very sensitive to false positives, low rates ($\sim 1 - 2\%$) of external colonisation and to a lesser extent to false negatives, as they all generate fake seed bank characteristic events, i.e., observations of plants in a patch which was previously empty along with its two neighbours. As this is likely to pose an issue when applying the estimator to real data, we introduced a variant of the BOA process for which the estimation is expected to be less sensitive to noise: the *noisy BOA process*, characterised by an additional noise parameter $\epsilon \in [0, 1]$. As illustrated in Fig 2, the main difference from the original BOA process is that after the extinction step (Step 2), each empty patch now contains plants with probability $\epsilon$. The goal of this additional parameter is to mitigate the effect of slight deviations from the model assumptions as well as to provide a buffer against potential false positives. This interpretation of $\varepsilon$ as a proxy for both the external colonization rate (assumed to be negligible) and the error rate when recording data (which can be expected to be very small, of order 0.01–0.05) has two consequences. First, we introduce an upper bound $\varepsilon_{max} \sim 0.1$ on the value of $\varepsilon$. Then, the parameter $\varepsilon$ has no direct biological interpretation, and is better seen as a tool to improve the estimation procedure. See S3 Text (Section 3) for a presentation of the estimator associated with the noisy BOA process, which now returns the posterior distribution $\mathbb{P}(s, p_{ext}, H, \epsilon|\text{Obs})$ of the four parameters $s$, $p_{ext}$, $H$ and $\epsilon$.

## Measuring the extinction risk

As discussed earlier, the key feature of the BOA process is the existence of a critical patch extinction probability $p_c(H)$ above which the metapopulation goes extinct and below which

it can survive and spread with nonzero probability. While we have no explicit expression for this survival probability, it increases when $p_{ext}$ decreases, and it is very small when $p_{ext}$ is close to the critical threshold. This observation suggests a simple way to assess the extinction risk of a metapopulation: compare the estimated value of $p_{ext}$ with the critical extinction threshold $p_c(H)$ for the estimated value of $H$. Our Bayesian framework makes it possible to include the uncertainty on the values of both $p_{ext}$ and $H$ when performing this comparison, leading to a first metric of the extinction risk called the *Global Extinction Risk*, or GER:

$$\text{GER} := \mathbb{P}(p_{ext} > p_c(H)|\text{Obs}) \tag{1}$$

$$= \sum_{h=0}^{H_{max}} \mathbb{P}(p_{ext} > p_c(H)|H = h, \text{Obs}) \times \mathbb{P}(H = h|\text{Obs}). \tag{2}$$

As the critical patch extinction probability $p_c(H)$ depends a lot on the value of the maximal dormancy duration $H$ (particularly for low to medium dormancy durations, see Fig 3), the performance of this metric is strongly correlated with the ability of the estimator to identify the correct value of $H$. However, as discussed earlier, the events that allow for the identification or rejection of a large value for $H$ are very infrequent, due to the somewhat restricted number of observation years (5 or 10 years). Unless the extinction probability is very high, models with a medium to large value for $H$ produce very similar outputs, so the probabilities $\mathbb{P}(H = h|\text{Obs})$ are expected to be extremely low below a certain value of $h$ (which depends on the observations), and then take relatively stable values. As the critical extinction threshold $p_c(H)$ increases with $H$, we expect the GER to underestimate the extinction risk for low to medium values of $H$ (that is, when the metapopulation is at risk of going extinct for intermediate values of $p_{ext}$) by weighting similarly medium and large values of $H$.

To avoid this issue, we introduce the following alternative metric for the extinction risk, called the *Maximal Global Extinction Risk*, or MaxGER. To do so, we first identify the lowest dormancy duration that is not rejected under a test with a 5% significance level, given the posterior distribution of $H$. That is, we define

$$H_{inf} := \min\{h \in [\![ 0, H_{max} ]\!] : \mathbb{P}(H \le h|\text{Obs}) \ge 0.05\}. \tag{3}$$

We then define the MaxGER as

$$\text{MaxGER} := \mathbb{P}(p_{ext} > p_c(H)|H = H_{inf}, \text{Obs}). \tag{4}$$

Contrary to the GER, the MaxGER does not weigh similarly models which are hard to distinguish given the size of the dataset, and only considers the lowest plausible value for $H$ (which corresponds to $H_{inf}$). Therefore, we expect the MaxGER to perform better than the GER for low to medium values of $H$, and to overestimate the extinction risk for large values of $H$ if $H_{inf}$ is significantly smaller than $H$. The choice of a standard 5% significance level was validated by a comparison to other possible significance levels, performed on simulated datasets (results not shown).

To choose between the MaxGER and the GER, we used the following protocol to compare the performance of the two metrics. For each parameter set listed in Table A in S3 Text, we simulated 30 BOA processes. We performed parameter inference under a BOA process, computed the average MaxGER and GER over the 30 simulated processes, and compared their values. We then chose the metric which fulfilled the following three conditions the best: 1) the extinction risk is close to 1 when $p_{ext}$ is above the critical extinction threshold $p_c(H)$, 2) the extinction risk is close to 0 when $p_{ext}$ is significantly smaller than the critical threshold,

and 3) the extinction risk becomes significantly different from 0 when $p_{ext}$ is below but close to the extinction threshold. This last condition is because the underlying BOA process has a significant probability of going extinct when $p_{ext}$ is close to the critical threshold, even when below it.

As the performance of the metrics depends on the quality of the estimation of $H$ and $p_{ext}$, which are both affected by the presence of noise in the data, we also assessed the performance of the MaxGER and GER metrics in the presence of false positives, false negatives, and low external colonisation rates. For each parameter set, noise type and noise intensity listed in Table B in S3 Text, we simulated 30 BOA processes. We performed parameter inference under a noisy BOA process with $\epsilon_{max} = 0.05$ or 0.1, and computed the average MaxGER and GER. To highlight how using the noisy BOA process can improve the estimation in this setting, we also performed parameter inference under the original non-noisy BOA process and compared the average MaxGERs and GERs to the ones measured under a noisy BOA process. We completed this study by computing the RMSE (*Root Mean Square Error*) on the estimation of $p_{ext}$ (see Fig P-R in S3 Text), along with the average value of $||H_{inf} - H||$ (see Fig S in S3 Text) and the distribution of the estimates of the MaxGER metric (see Fig I-L in S3 Text).

## Analysis of the *Paris 12* dataset

In order to assess the extinction risk of the 181 metapopulations listed in Table B in S1 Text, we assumed that the plant dynamics was described by a noisy BOA process whose maximal dormancy duration $H$ and noisy intensity $\epsilon$ were uniform across streets for a given species. In particular, in line with previous studies carried out on urban tree bases using SPOMs [5, 8] and in line with information available in the LEDA database [23], we assumed that the number of seeds produced by each plant was large enough for the presence/absence approximation to be valid. The death of all plants at the end of each generation in the noisy BOA process is consistent with the observation that in urban tree bases, standing vegetation is removed yearly by gardeners as part of the tree bases management. As survival is equivalent to spreading through the set of patches for the BOA process, we interpreted low values of the MaxGER as indicative of the species ability to use the corresponding portion of street as an ecological corridor.

For each plant species, we ran the estimation simultaneously over all streets listed in Table B in S1 Text, and allowed the extinction probability to vary from one street to another. See S2 Text (Section 4) for a more detailed presentation of the estimation method and S3 Text (Section 4) for an illustration of how grouping streets improves the estimation.

To provide an overview of the variation of the extinction risks and patch extinction probabilities across species and streets, we performed a repeatability analysis on the MaxGER and the average of the posterior distribution of the patch extinction probability (thereafter referred to as *Local Extinction Risk*, or LER) using the rptR package [24]. The repeatability analysis was based on 1000 parametric bootstraps, and we assessed its statistical significance by a likelihood ratio test comparing the model including grouping factors (the species or the portion of street) and one excluding it.

As we expect different portions of the same street to have closer patch extinction probabilities than portions of different streets, we compared the posterior distribution of $p_{ext}$ across different portions of streets for a same species, by computing the *Standardised Mean Difference* (or SMD) for each pair of portions of streets as follows. If $\overline{p_{ext,1}}$, $\overline{p_{ext,2}}$, $\sigma_{ext,1}$ and $\sigma_{ext,2}$ are the mean and standard deviation of the posterior distribution of $p_{ext}$ for two different portions of

**Table 1. Species traits considered as part of the study, and statistical tests used to assess their effect on the local and global extinction risks (averaged over all streets for each species).** The database used to identify these traits were the LEDA database [23] and BaseFlor [25].

| Plant trait | Values taken | Statistical test |
|---|---|---|
| Dispersal mechanism | Anemochorous (by wind) Barochorous (by gravity) Epizoochorous (by animals, without ingestion) Autochorous (mechanical action by the plant itself) | Kruskal-Wallis test, Dunn's test |
| Flowering duration | Short ($\leq 3$ months) Medium ($\geq 4$ and $\leq 6$ months) Long ($\geq 7$ months) | Kruskal-Wallis test, Dunn's test |
| Seed mass | Quantitative variable (in grams) | Spearman's rank correlation test |
| Heat preference | Sensitive (Ellenberg value $\leq 6$) Resistance (Ellenberg value $\geq 7$) | Wilcoxon test |
| Pollination vector | Insect Wind Selfed | Kruskal-Wallis test, Dunn's test |
| Maximal height | Quantitative variable (in centimeters) | Spearman's rank correlation test |
| Beginning of flowering period | Early (in April or earlier) Late (in May or later) | Wilcoxon test |

streets, the SMD is equal to

$$SMD = \frac{|\overline{p_{ext,1}} - \overline{p_{ext,2}}|}{\sqrt{\frac{\sigma_{ext,1}^2}{2} + \frac{\sigma_{ext,2}^2}{2}}}. \tag{5}$$

For each species and for each street, we computed the average SMD of the posterior distribution of $p_{ext}$ computed in two portions of this street and compared it to the average SMD for a portion of this street and a portion of a different street.

In order to understand the drivers of the extinction risk at the patch and metapopulation scale, we first used a linear regression model with the MaxGER or LER as the response variable, to test whether they depended on the species or the portion of street. We then tested the effect of seven different species traits on the MaxGER and LER metrics, using nonparametric statistical tests as the data did not follow a normal distribution (Kruskal-Wallis or Wilcoxon test for qualitative variables and Spearman's rank correlation test for quantitative variables). The specific traits we studied, which are listed in Table 1, are often used in ecology and conservation biology due to their direct or indirect influence on survival or reproduction, are related to the characteristics of the urban environment (e.g. the heat preference), or may reduce the risk of the plant being killed by gardeners before producing seeds (maximal height, flowering duration, beginning of flowering period). Therefore, they express the ability of plants to colonise new habitats or avoid local extinction in urban environments, and they may influence the long-term survival of the plant metapopulation.

## Results

### Testing the assessment of the extinction risk using simulated data

The analysis of the estimation results on simulated BOA and noisy BOA processes showed that the overall behaviour of the MaxGER and GER metrics is the one required of a good

metric of the extinction risk. Indeed, both are close to zero when $p_{ext}$ is below the critical threshold, nonzero above the critical threshold, and start increasing before the critical threshold for $p_{ext}$. The maximum value of the extinction risk increased with the number of patches $N$ and the monitoring duration $T$, and decreased with the noise intensity $\varepsilon$. See S3 Text (Section 2.1) for an illustration of the effect of $N$, $T$ and $\varepsilon$ on the assessment of the GER metric, and S3 Text (Section 2.2) for an illustration of their effect on the assessment of the MaxGER metric.

Although the behaviour of the MaxGER and GER metrics were very similar, some differences are visible, especially in the case $T = 5$ and when the noise intensity is high. The complete results can be found in S3 Text (Section 2), and show that in many instances (see e.g. $(N, T)$ = (50, 5), $(N, T)$ = (100, 5)), the MaxGER metric starts increasing earlier than the GER metric and reaches higher values. Therefore, the MaxGER metric better fulfils the third condition required of a good metric of the extinction risk ("the extinction risk becomes significantly different from 0 when $p_{ext}$ is below but close to the extinction threshold"), and the rest of the section focuses on the performances of the MaxGER metric.

The introduction of false negatives did not significantly affect the assessment of the MaxGER metric or the estimation of $p_{ext}$. The introduction of false positives or external colonization was mostly visible when $H = 0$ or $H = 1$ and for parameter inference under a BOA process (without noise): the extinction risk was incorrectly assessed as very low or null for extinction probabilities above the critical threshold. See S3 Text (Section 2.3) for the complete results.

The performances of the estimations of $p_{ext}$ and $H$ on non-corrupted datasets were similar under BOA and noisy BOA processes. The effect on the estimation of $p_{ext}$ of the introduction of false negatives/positives or external colonization was mostly visible when $H = 0$ and for inference under a BOA process without noise. Furthermore, in the case of false positives and external colonization, it also led to the incorrect identification of a long dormancy duration when $H = 0$ or 1, again for inference under a BOA process without noise. See S3 Text (Section 3) for the complete results.

## Global extinction risk

In line with the results of the simulation study, we used the MaxGER metric to assess the extinction risk of a species at the street scale. Its application to the *Paris 12* dataset revealed contrasting extinction risks between metapopulations (see Fig 4A). While 113 out of 181 pairs had a null extinction risk, a MaxGER greater than 0.05 was measured in 61 pairs, and a MaxGER greater than 0.50 in 42 pairs. The regression model identified nine species associated with a higher global extinction risk, the largest values corresponding to *Plantago lanceolata*, *Senecio inaequidens* and *Lactuca serriola*. Conversely, the extinction risk did not differ significantly from one street to another. The repeatability of the MaxGER between streets for a given species was very high (repeatability $R = 0.769 \pm 0.067$, p-value $< 10^{-5}$), while the repeatability between species for a given portion of street was null. See complete estimation results in Table C in S1 Text.

Three plant traits were identified as having a significant effect on the global extinction risk when accounting for multiple testing: the maximal height ($p - value = 0.0048$), whether the flowering period begins early (in April or earlier, $p - value = 0.0092$), and the duration of the flowering period ($p - value = 0.0027$). A high maximal height was associated with a higher average MaxGER. So was a flowering period of intermediate duration (4 to 6 months) or starting in May or later. Complete estimation results can be found in Fig 5 and in Table D in S1 Text.

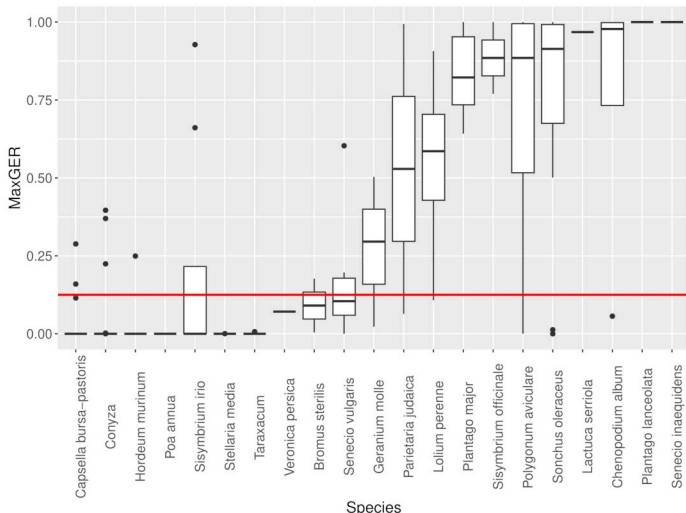

**(A)** Distribution of MaxGERs across streets for each species. The red line indicates the value MaxGER= 0.125 below which the extinction risk at the metapopulation scale is considered as low.

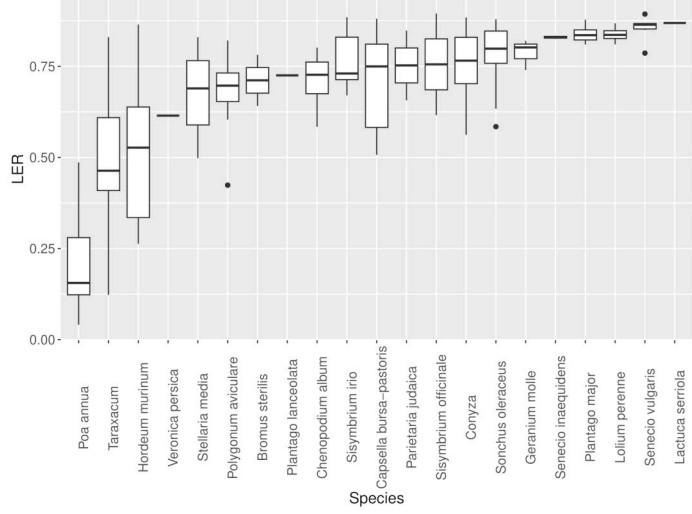

**(B)** Distribution of LERs across streets for each species.

**Fig 4. Global (MaxGER) and local (LER) extinction risks of each species listed in Table B in S1 Text, across portions of streets listed in Table B in S1 Text.**

## Local extinction risk

We recall that the LER, or *Local Extinction Risk*, is the average of the posterior distribution of the patch extinction probability $p_{ext}$. The analysis showed that local extinction risks are very high overall (see Fig 4B). Only 4 species had a LER below the threshold for survival without a seed bank ($p_c(0) = 0.461$) in at least one portion of street: *Poa annua* (21 out of 22 pairs), *Taraxacum sp.* (9 out of 19 pairs), *Hordeum murinum* (7 out of 16 pairs) and *Polygonum aviculare* (1 out of 11 pairs). The repeatability of the LER between streets for a given species was high (repeatability $R = 0.621 \pm 0.092$, p-value $< 10^{-5}$), while the repeatability between species for a given portion of street was very low (but significantly different from zero: $R = 0.038 \pm 0.026$,

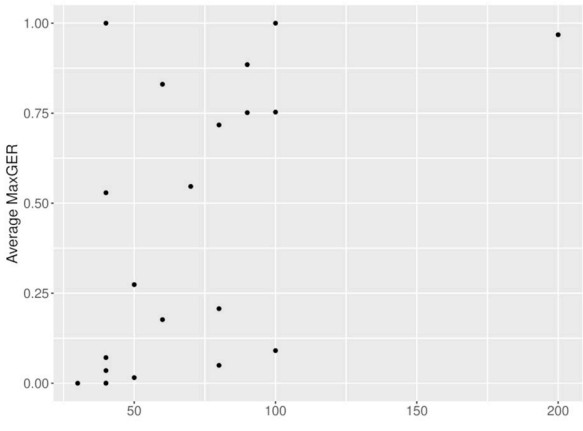

**(A)** Maximal height

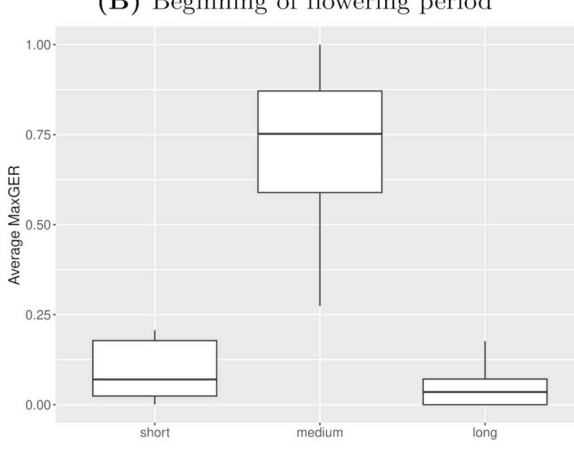

**(B)** Beginning of flowering period

**(C)** Flowering duration

**Fig 5.** Illustration of the relation between the global extinction risk (as quantified by the MaxGER metric, and averaged over all streets for each species) and three traits for which a significant correlation was identified: (a) the maximal height (p-value = 0.004763), (b) the beginning of the flowering period (p-value = 0.00915) and (c) the flowering duration (p-value = 0.002677).

p-value = 0.032). Comparing the mean SMDs between portions of a same street or of different streets showed that patch extinction probabilities were on average closer inside a street than between streets, for all but one species and one portion of street (see Fig C in S1 Text).

Three species were identified as associated with a lower local extinction risk, while three portions of streets were associated with a higher LER (two of which belonging to the same street). See estimation results in Table E in S1 Text. When accounting for multiple testing using the Holm-Bonferroni procedure, no plant trait was identified as having a significant effect on the local extinction risk, though the correlation was borderline significant for two of them (the maximal height and the beginning of the flowering period). Complete estimation results can be found in Table F and Fig B in S1 Text.

### Presence of a seed bank

The presence of a seed bank ($\mathbb{P}(H \geq 1|\text{Obs}) \geq 0.95$) was identified in 19 out of 21 species, while the absence of a seed bank ($\mathbb{P}(H = 0|\text{Obs}) \geq 0.95$) was only identified in one species (*Plantago lanceolata*). A medium to long dormancy duration ($H_{inf} \geq 2$) was identified in 15 species. Complete estimation results can be found in Table G in S1 Text.

## Discussion

This study introduces a new approach to assess the extinction risk of a plant metapopulation in a fragmented landscape. Our approach is based on the MaxGER metric, which can be computed using presence-absence data of standing vegetation and takes into account the potential presence of a seed bank. The application of the MaxGER metric to a dataset based on a long-term survey of the spontaneous flora in urban tree bases across Paris (*Paris* 12 dataset, [12]) produced the first assessment of the extinction risk of 21 plant species commonly found in urban tree bases. Overall, our results highlight to what extent urban tree bases can act as ecological corridors and can be of interest to gardeners to design weeds management strategies.

### Extinction risk of plant metapopulations in urban tree bases across Paris

The analysis of the *Paris* 12 dataset uncovered a set of 10 core species which can survive in urban tree bases without external colonisation, which we interpret as having an average MaxGER below 0.125. These species, listed in Table 2, are able to use tree bases as ecological corridors to spread in an urban environment, while others must rely on external colonisation to

**Table 2. Species identified as having a low or high global extinction risk, as assessed by the MaxGER metric.** We interpreted an average MaxGER $\leq 0.125$ across streets as having a low extinction risk. Species listed as having a high global extinction risk are those identified as part of the regression analysis.

| Low global extinction risk | High global extinction risk |
|---|---|
| *Bromus sterilis* | *Chenopodium album* |
| *Capsella bursa-pastoris* | *Lactuca serriola* |
| *Conyza sp.* | *Lolium perenne* |
| *Hordeum murinum* | *Plantago lanceolata* |
| *Poa annua* | *Plantago major* |
| *Senecio vulgaris* | *Polygonum aviculare* |
| *Sisymbrium irio* | *Senecio inaequidens* |
| *Stellaria media* | *Sisymbrium officinale* |
| *Taraxacum sp.* | *Sonchus oleraceus* |
| *Veronica persica* | |

balance local extinctions and survive in urban tree bases. In particular, this is the case for the nine species identified as associated to a higher global extinction risk (see list in Table 2).

By design of the MaxGER metric, two factors can contribute directly to a low global extinction risk: a low patch extinction probability (or local extinction risk, measured by the LER metric and potentially influenced by plant traits), and the ability of the species to form a seed bank. It turned out that in urban tree bases, the main driver of the global extinction risk is the maximal dormancy duration $H$. Indeed, local extinction risks were overall very high, and in the vast majority of cases above the critical threshold for survival without a seed bank or with a short-term seed bank ($H = 1$). A high maximal dormancy duration makes it possible for species to survive despite extremely high local extinction risks (see e.g. *Senecio vulgaris*, which has the second highest average LER but low MaxGER values). Conversely, low maximal dormancy durations can result in high global extinction risks despite lower-than-average LERs. Two prime illustrations of this are the species *Polygonum aviculare* ($H_{inf} = 1$) and *Plantago lanceolata* ($H = 0$): these two species have relatively low LERs compared to other species, but their LER is still too high to survive without a long-term seed bank. Therefore, our study gives an experimental identification of an environment in which the formation of a seed bank is necessary for the survival of a plant metapopulation, and gives further experimental evidence that seed banks can contribute to survival in disturbed environments (see also [26]).

Our study uncovered other traits influencing the extinction risk at the metapopulation scale: the maximal height, the beginning of the flowering period and the length of the flowering period. The first two are in line with the expected effect of the annual clearing of vegetation by gardeners. Indeed, smaller species are more likely not to be cleared, and species that start their flowering period late run the risk of being weeded before having produced seeds. It is worth noting that the correlation is only significant at the metapopulation scale (as measured by the global extinction risk) and not at the local scale (as measured by the local extinction risk). This may be further evidence of the fact that survival at the metapopulation scale or at the local scale are not interchangeable.

The effect of the length of the flowering period on the global extinction risk (higher risk when the flowering period has an intermediate length) may seem unexpected at first. A possible explanation is that it is a consequence of correlations between the different plant traits tested, which we could not account for with nonparametric statistical tests. Indeed, the flowering duration is strongly correlated with whether the flowering period begins early or late, and most species with a flowering period of intermediate length start blooming in May or later.

The three traits that we identified as influencing the extinction risk are therefore related to the action of gardeners on urban tree bases. Conversely, while species with a high Ellenberg value could be expected to be more adapted to urban environments, we did not find an effect of the heat preference on the extinction risk. Similarly, we did not find a correlation between the dispersal mechanism (which is linked to the dispersal capacity) and the extinction risk. As explained above, the expected effect of the dispersal mechanism on survival in urban tree bases is unclear: while strong dispersal capacities would allow to avoid local extinctions, there is evidence of selection against long-range dispersal in urban tree bases, due to the high fragmentation [9].

Another possible factor that influences the extinction risk is the portion of street considered. The analysis of the *Standardised Mean Differences* of the estimated patch extinction probabilities uncovered that the local extinction risk is somewhat consistent between portions of a same street for each species. However, we did not identify an effect of the portion of street considered on the MaxGER metric and only identified a limited effect on the LER metric. Our results are in stark contrast with previous studies on Paris urban tree bases [8, 10], which uncovered an effect of the street features on a species abundance and dynamics (but see [11],

in which a similar result was obtained for an equivalent of the local extinction risk). A first possible explanation is that the studies performed in [8, 10] did not consider the potential presence of a seed bank when performing parameter inference. Conversely, our study and the one performed in [11] both took into account the potential presence of a seed bank, and none uncovered a strong effect of the street on the observed dynamics. Another possible explanation is that we only considered portions of streets in which the focal species was abundant enough. By excluding portions of streets into which the species is extinct or close to extinction, we may hide the effect of the street on the extinction risk and hinder its detection.

In order to assess the validity of our estimates, we used the LEDA database [23] to check whether the presence of a seed bank had already been documented in the literature for each of the species considered as part of the study. Out of the 18 species for which seed bank longevity data was available in the LEDA database, the ability to form a long-term seed bank (interpreted as $H \geq 5$ in the LEDA database) was documented for 16 species, and the ability to form a short-term seed bank ($1 \leq H \leq 4$) was documented for one species. The records available in the LEDA database also highlighted that dormancy durations are highly dependant on the environment. A key example of this is *Plantago lanceolata*, for which maximal dormancy durations reported in the LEDA database ranged from $H = 0$ to $H \geq 5$. This gives evidence of the need for indirect approaches to assess the ability to form a seed bank in the environment of interest (whose conditions might not be easily replicable in a laboratory environment), for instance using presence/absence data (see also [11, 17, 19]) or genomic data (see e.g. [27]).

The main limit of our study is that we only considered colonisation from neighbouring patches and low rates of colonisation from an external source. These choices were motivated by evidence of selection for low-distance dispersal in such an environment [9], and by the need to control external colonisation rates to assess the extinction risk in the absence of external colonisation. However, it may be the case that whether colonisation from neighbouring patches is predominant or negligible compared to colonisation from an external source depends on the streets considered, as demonstrated for *Crepis sancta* in [5]. Despite this, the study of the *Standardised Mean Differences* of patch extinction probabilities between portions of the same street or of different streets suggests that our modelling framework is adapted to the *Paris* 12 dataset. Indeed, for all but one street, the patch extinction probability of a species in a portion of this street was on average closer to the patch extinction probability in another portion of this street than in a portion of a different street (see Fig C in S1 Text). In other words, the model does capture similar dynamics in portions of the same street.

Another limit of this study is that due to the lack of metacommunity models with a seed bank component, we had to neglect interactions between species and consider each species as forming an independent metapopulation. However, this assumption is likely to only be true to some extend due to the limited size of a tree base, which may induce competition for resources. The lack of SPOM metacommunity models (with or without a seed bank component) has recently been identified as an important future area of research [14], and our study highlights the need to take into account the potential presence of a seed bank when developing such models.

## Assessing the extinction risk of a plant metapopulation

In this study, we introduced a new Bayesian framework for parameter inference under a BOA process. It completes existing estimation procedures for *Stochastic Patch Occupancy Models* [13, 17, 19] by allowing parameter inference when colonization occurs primarily between neighbouring patches and when a seed bank is potentially present. The approach we used to interpret the results of the estimation procedure mirrors the one introduced in [11], in the

sense that we focused on the effect of the process on the observed dynamics rather than on the process itself. In particular, we do not try to differentiate parameter sets giving rise to similar observed dynamics, which mechanically improves the performances of the estimation procedure.

The mathematical properties satisfied by the BOA process [18] allowed us to derive two possible metrics assessing the extinction risk of the metapopulation (the GER and MaxGER metrics). The interest of this approach is that the fact that these metrics do measure the extinction risk is a direct consequence of the mathematical results in [18], which makes it unnecessary to test this using numerical simulations. As the computation of these two extinction risk metrics depends on the values of the critical patch extinction probabilities $(p_c(H))_{H \geq 0}$, which in turn depend on the geometry of the system studied, our estimation procedure cannot be applied directly to metapopulations in which patches are not equidistant and along a line. However, it is possible to obtain precise numerical approximations of $(p_c(H))_{H \geq 0}$ for any kind of spatial structure (generalising the approach used in [18]), which would allow one to apply our approach to more general metapopulations.

The numerical exploration of the performances of the MaxGER metric showed that the metric is well assessed in a wide variety of biologically relevant situations, including low but nonzero rates of errors when recording data. This last observation is particularly important for the applicability of the approach, since it implies it can be used on data coming from citizen science programs, for which a standardised protocol and good training methods make it possible to achieve error rates of less than 0.05 [28, 29]. Moreover, while it was not the primary focus of the study, the numerical exploration showed that the patch extinction probability $p_{ext}$ is also well estimated in biologically relevant situations, allowing us to complete our study of the global extinction risk (at the street scale) by a study of the local extinction risk (at the patch scale). The application to the *Paris* 12 dataset highlighted that these two studies are complementary and that measuring the local extinction risk can shed light on how a species survives at the metapopulation scale. Notice that while the patch extinction probability $p_{ext}$ was estimated well, this was not the case of all the parameters, and in particular of the maximal dormancy duration $H$. Therefore, the MaxGER metric can be more informative than the unprocessed estimates of all the parameters of the process.

The main limit of our assessment procedure for the extinction risk is that both the GER and MaxGER metrics depend on the quality of the estimation of the model parameters in a peculiar way, in the sense that the quality of the assessment of the extinction risk is not a monotonic function of the quality of the estimation of the model parameters. Indeed, we saw earlier that the BOA process has a significant probability of going extinct when $p_{ext}$ is below but close to the critical threshold for survival. As the GER and MaxGER metrics are computed by looking at the posterior probability that $p_{ext}$ is above the critical threshold $p_c(H)$, we need the variance of the posterior distribution to be large enough to ensure that patch extinction probabilities close to the critical threshold are associated to a significant extinction risk. Our numerical exploration of the performance of the estimation procedure showed that this condition was always satisfied in biologically relevant situations.

## Conclusion

We introduced a new Bayesian framework for parameter inference in plant metapopulations characterised by colonisation between neighbours and the potential presence of a seed bank, and developed a metric assessing the extinction risk of a plant metapopulation using presence/absence data. The application of our estimation procedure to a dataset of urban tree bases illustrated how it can be used to disentangle persistence at the local and global scales, and

understand their drivers. Furthermore, this study represents the first assessment of the extinction risk of the common herbaceous flora of Paris urban tree bases. It identifies which plant traits make it possible to survive autonomously in urban tree bases, and which species are only observed due to a constant influx of new seeds from neighbouring tree bases. These results have important implications with respect to plant dynamics in an urban environment and can be used to design improved management strategies for urban tree bases.

## Supporting information

**S1 Text. The Paris 12 dataset—Presentation and estimation results.** Additional information regarding the Paris 12 dataset (including a map of the study area and a list of the species and streets taken into account as part of the study, see Tables A, B and Fig A) as well as the detailed results of the statistical tests performed. The list of these statistical tests can be found in the section "Material and Methods—Analysis of the *Paris* 12 dataset" of the manuscript, and a summary of these results in the "Results" section (subsections "Global extinction risk", "Local extinction risk" and "Presence of a seed bank").
(PDF)

**S2 Text. The BOA Process: Mathematical formulation and estimation procedure.** Presentation of the framework implemented to perform inference under a BOA process. The corresponding code can be found online at https://github.com/cmantoux/boa-process, along with detailed explanations regarding how to reproduce the results presented in this article.
(PDF)

**S3 Text. Assessment of the performances of the estimation procedure.** Detailed results of the assessment of the performance of the MaxGER and GER metrics, as described in the section "Material and Methods—Measuring the extinction risk" of the manuscript.
(PDF)

## Acknowledgments

The authors thank Ugo Sinacola, who contributed to the statistical analyses of the estimated extinction risks of plants in urban tree bases. They also thank Mona Omar, Noëlie Maurel and Florence Devers who organised and carried out the floristic survey from 2009 to 2018.

## Author Contributions

**Conceptualization:** Apolline Louvet, Clément Mantoux, Nathalie Machon.

**Data curation:** Apolline Louvet.

**Formal analysis:** Apolline Louvet, Clément Mantoux.

**Methodology:** Apolline Louvet, Clément Mantoux.

**Software:** Clément Mantoux.

**Supervision:** Nathalie Machon.

**Validation:** Apolline Louvet.

**Visualization:** Apolline Louvet, Clément Mantoux.

**Writing – original draft:** Apolline Louvet.

**Writing – review & editing:** Apolline Louvet, Clément Mantoux, Nathalie Machon.

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
