## [Decision Letter · Decision Letter 0]

16 Feb 2024

Dear Louvet,

Thank you very much for submitting your manuscript "Assessing the extinction risk of the spontaneous flora in urban tree bases" for consideration at PLOS Computational Biology.

As with all papers reviewed by the journal, your manuscript was reviewed by members of the editorial board and by several independent reviewers. In light of the reviews (below this email), we would like to invite the resubmission of a revised version incorporating the comments and suggestions from the referees. 

We cannot make any decision about publication until we have seen the revised manuscript and your response to the reviewers' comments. Your revised manuscript is also likely to be sent to reviewers for further evaluation.

Sincerely,

Jere Koskela

Guest Editor

PLOS Computational Biology

Denise Kühnert

Section Editor

PLOS Computational Biology

Editor comments:

The paper has been reviewed by three referees. All agree that the work is interesting, that the methodology is both rigorous and novel, and that the paper would be a good fit for PLOS Computational Biology in principle. However, all three also request some clarifications, and in one case, further validation of the robustness of the extinction risk estimation step in the methodological pipeline. One referee also noted that the reproducibility of the work should be improved by sharing the code used to produce these results in a format that others can easily rerun.

I'm inclined to agree with the referees on all counts. This manuscript presents novel and interesting work on a topic which is well-suited to PLOS Computational Biology. I would welcome a revised version incorporating the comments and suggestions from the referees.

Reviewer's Responses to Questions

**Comments to the Authors:**

Reviewer #1: uploaded as an attachment.

Reviewer #2: The authors assess the extinction risk of various plant species growing in urban tree bases.

For this, the authors develop a Bayesian parameter inference method based on a (noisy) stochastic occupancy model (as a hidden Markov model, HMM), with the innovation of a dormancy component.

They also introduce two measures of measure the extinction risk.

They apply this method to a data set collected in Paris, estimate extinction risks for the observed species and provide a second step analysis based on the measured extinction risks to assess whether certain biological properties lead to significantly different extinction risk.

In general, I find the methods suitable for the tasks at hand.

The authors argue convincingly that adding noise to the HMM will allow to capture unmodelled influences, and are honestly evaluating strengths and weaknesses of their approach. The estimation methods are sound from my viewpoint. The approach is interesting and applicable for other data sets, also in other contexts.

The only criticism I have is the second step analysis of evaluating which biological factors influence the extinction risk.

As the authors show that their estimates tend to be noisy, it would be nice to show that these second level tests are working as intended

I would suggest to use the simulation approach the authors use to assess how different the distributions of extinction risks measures are for different values of the (true) extinction probability (and the other parameters) to assess the power and size of these tests.

As a possible biological assessment of the validity of their estimates, I encourage the authors to provide a short review of whether dormancy estimates for the species covered are known from germination experiments etc, and, if there are, compare these with their H estimates.

Additionally, here is a list with suggestions for small changes and some small questions:

-p.5, l.89-: Please provide an explanation why your model that does not track the number of seeds makes sense biologically (essentially, that you assume the plants provide enough seeds). Is this justified for the species you consider?

-p.7, l.124: Describe what is estimated if the true H>10.

-p.8, l.133: Can you omit the \\in\\mathbb{N} here?

-p.10, Eq. 3: Give a rationale/heuristic of your choice of 0.05 as threshold

-p.13, l.258: Please refer to an appropriate table/figure here

Reviewer #3: Overview: This paper analyzes a large, public vegetation database collected from city streets in Paris. The paper describes how streets were partitioned to look at the data through the lens of “metapopulation” theory, which considers movement of propagules in space (neighbors) and time via a seed bank. The paper develops a Best Occupancy Achievable (BOA) process to account for sampling and presence-absence data of different species with the goal of quantifying extinction risk. The paper identifies plant traits that may be associated with increased extinction risks, which has implications for the management of urban ecosystems.

Major issues: There are a number of things that I like about this paper. For example, I think the project is creative in its use of a large, one-of-a-kind database. The paper is quantitatively rigorous and develops some novel statistical approaches for quantifying extinction risk from presence-absence data. And conceptually, I liked that the paper integrates ecological ideas related to metapopulations (including dispersal, dormancy, and patches) into the work. That being said, I have some thoughts and recommendations that the authors may want to consider:

1. At times, I found myself confused by the organization and motivation of the paper, especially after reading the Abstract and Introduction. The paper starts off talking about “tree bases”. I suspect that this is not something that is going to immediately resonate with most readers, especially those of PLOS Computational Biology. Similarly, the notion or importance of corridors in the context of conservation biology and urban ecology could be better developed for readers. It seems to me that the paper is mostly motivated by the development of the extinction risk metrics from presence-absence data, and that perhaps this is what makes the research most relevant to PLOS Computational biology. A major portion of the paper is dedicated to describing these methods. While the plant data set is very interesting, is perhaps the novel quantitative angle the development of the BOA process? I probably would have benefited a bit more from some background on the existing models that are used for estimating extinction risks, and what the challenges or needs are for estimating extinction risk in such a study like this one, which I assume has to wrestle with presence-absence data and some inferred dynamics (seed banks and small-scale dispersal).

2. Perhaps this doesn’t change how the data are analyzed, but “metapopulation” studies tend to focus on a single species, while “metacommunities” study the spatial processes regulating the composition of multispecies assemblages, and often consider processes like dispersal and dormancy on dynamics, including extinction. Is there a reason why the paper is focused on metapopulations and not metacommunities? Is it reasonable to assume that the species in a tree base are not interaction with one another? In other words, are extinction risks not affected by the presence-absence of other species either at the local or regional scale? I think this is an important question --- or assumption --- that needs to be addressed and justified in some way.

3. The paper talks about traits. It seems a bit awkward to repeatedly say “biological traits” in a paper that is about plants, as there are no other traits that would be relevant to the study. It is interesting that flowering time, maximum height, and flowering duration are related to extinction risk. The discussion of these traits is somewhat underdeveloped. I was also curious about the associated statistical analyses. Flowering period and flowering duration are treated categorically. What type of correlation was performed? And why were only p-values reported and not correlation coefficients, which would inform reader about the strength of the association? How are these traits related to dispersal or seed banks?

4. I am a little confused about the support (i.e., data) for the conclusion that tree bases act as ecological corridors. Which data are used to bolster this inference? Perhaps this could be made clearer?

5. There is a lot of data in the Supplementary materials. I did not spend an extended amount of time going through this section, but I did note that it contrasts with the relatively small amount of data presented in the main manuscript, which effectively reports (three) trait correlations (Fig. 3) and description of extinction risks for 21 plant species (Fig. 4).

6. The paper seems to invoke an importance of seed banks, but in the Results section, all we’re really told is that 19 out of 21 species had a seed bank. It seems that species without seed banks have higher estimated extinction probabilities? Maybe this point could be better emphasized.

Minor issues:

Maybe the paper should define what tree bases are, and perhaps include a picture as Fig. 1

Abstract: “…some of them being not previously known” vague reference

Abstract: “…approach we introduced” vague reference

Author summary: “we introduced a new method” vague reference.

Author summary: “Our approach takes into account several factors previously identified…” vague reference.

Author summary: “We applied our approach” vague and hard to follow at this stage of paper.

Line 28: what is meant by “spontaneous flora”?

Line 55: is this a “working hypothesis” (that will be tested) or is it an assumption?

Line 60 (and throughout): “pair species/portion” is sort of awkward and not well defined.

Line 143: what are “fake seed bank characteristic events”?

Line 173: “that allow to identify” fix grammar

Table 1: I’m not familiar with the dispersal mechanisms described in the “values taken” column

**Have the authors made all data and (if applicable) computational code underlying the findings in their manuscript fully available?**

Reviewer #1: Yes

Reviewer #2: Yes

Reviewer #3: **No: **I think the data is publicly available, but it might be good for them to make accessible the data they used along with code, which I don't see any reference to (e.g., github, zenodo, etc.)

PLOS authors have the option to publish the peer review history of their article (what does this mean?). If published, this will include your full peer review and any attached files.

Reviewer #1: No

Reviewer #2: No

Reviewer #3: No
---

## [Decision Letter · Decision Letter 1]

24 May 2024

Dear Louvet,

We are pleased to inform you that your manuscript 'Assessing the extinction risk of the spontaneous flora in urban tree bases' has been provisionally accepted for publication in PLOS Computational Biology.

All reviewers agree that the paper is essentially ready for publication, and we concur. One referee has made some small suggestions for improving the clarity of the abstract along the lines already done for the introduction. We would recommend implementing their comments and submitting a (slightly) revised manuscript for publication. Either way, we see no reason for another round of review. Congratulations on a very nice piece of work!

Best regards,

Jere Koskela

Guest Editor

PLOS Computational Biology

Denise Kühnert

Section Editor

PLOS Computational Biology

Reviewer's Responses to Questions

**Comments to the Authors:**

Reviewer #1: I am pleased with the modifications made by the authors, which address all the points I had raised about the initial submission.

In particular, I appreciate the extension of the discussion regarding the significance of seed banks for survival in urban tree bases, which addresses a major concern from the previous submission.

Furthermore, the modifications made to the figures have effectively resolved the issues previously raised.

These revisions have enhanced the clarity and impact of the manuscript. I fully support the publication of this work in this prestigious journal.

I also take this opportunity to congratulate the authors on their solid contribution.

Reviewer #2: I enjoyed reading the new version - many thanks for your work on this. All my previous suggestions have been dealt with in the new version and I have no further suggestions for improvement.

Reviewer #3: The authors have done a good job of revising the paper. Much of my original confusion has been cleared up. Most people who come across the publication will first read the Abstract. Unlike the Introduction, which is now very nicely set up, the first few sentences of Abstract are written in a confusing manner, in my opinion. I would recommend that the editor give the authors the opportunity to slightly revise.

First sentence: "As the design of tree bases along streets in cities makes them potential ecological corridors, urban tree bases may be a key contributor to the overall connectivity of the urban ecosystem."

-- The topic sentence above is very passive. First, the paper is not really about _design_ per se, which is the subject of the sentence. Second, "tree bases" are not something that most readers -- even conservation biologists -- will be familiar with. (the Introduction, by contrast, does a nice job of explaining and providing context).

Second sentence "However, they are also a highly fragmented environment".

-- Grammatically, this sentence is ambiguous. For example, is it referring to the subject of the previous sentence, i.e., the "design of tree bases"?

Third sentence: "The goal of this study was to assess the plant species ability"

-- "species" should be changed to "species' " (possessive)

First sentence of author summary: "Understanding how biodiversity is maintained in a urban environment is an important

question in urban ecology"

-- change "urban ecology" to "ecology"

**Have the authors made all data and (if applicable) computational code underlying the findings in their manuscript fully available?**

Reviewer #1: Yes

Reviewer #2: Yes

Reviewer #3: None

PLOS authors have the option to publish the peer review history of their article (what does this mean?). If published, this will include your full peer review and any attached files.

Reviewer #1: No

Reviewer #2: No

Reviewer #3: No

---

## [Editor Report · Acceptance letter]

21 Jun 2024

PCOMPBIOL-D-23-01839R1 

Assessing the extinction risk of the spontaneous flora in urban tree bases

Dear Dr Louvet,

I am pleased to inform you that your manuscript has been formally accepted for publication in PLOS Computational Biology. Your manuscript is now with our production department and you will be notified of the publication date in due course.

With kind regards,

Olena Szabo
